# Application of a Novel Formulation of 1-Aminocyclopropane-1-carboxylic Acid (ACC) to Increase the Anthocyanins Concentration in Table Grape Berries

**DOI:** 10.3390/plants14071058

**Published:** 2025-03-29

**Authors:** Aline Cristina de Aguiar, Danielle Mieko Sakai, Bianca Liriel Martins Barbosa, Stefanie do Prado da Silva, Fábio Yamashita, Sergio Ruffo Roberto

**Affiliations:** Agricultural Research Center, Agronomy Department, State University of Londrina, Celso Garcia Cid Road, Km 380, Londrina 86057-970, Brazil; alinecristina.aguiar@uel.br (A.C.d.A.); danielle.sakai@uel.br (D.M.S.); stefanie.prado.tadeu@uel.br (B.L.M.B.); bianca.liriel.martins@uel.br (S.d.P.d.S.); fabioy@uel.br (F.Y.)

**Keywords:** anthocyanins, bioactive compounds, quality attributes, skin color, *Vitis vinifera* L.

## Abstract

The objective of this work was to assess different concentrations of a novel formulation of 1-aminocyclopropane-1-carboxylic acid (ACC) on anthocyanin accumulation and color development, as well as on the physicochemical characteristics of the ‘Benitaka’ table grape grown in a subtropical region in two application forms. The trial was conducted on a commercial property located in a subtropical area in Brazil in 2022. Treatments included different concentrations of a new formulation containing 400 g kg^−1^ of ACC, ranging from 0 to 125 g 100 L^−1^, as well as a standard concentration of a formulation containing 100 g L^−1^ of abscisic acid (*S*-ABA): 3.2 L ha^−1^. The exogenous application of ACC was performed at the beginning of berry ripening (véraison), while that of *S*-ABA was performed twice: the first, at véraison, and the second, 7 days later. The concentration of total anthocyanins, berry color index, physicochemical characteristics, and sensory–visual analysis of color coverage of the bunches were evaluated weekly, while berry firmness was appraised at harvest. A single exogenous application of ACC or two applications of *S*-ABA resulted in daily increment rates that provided a high accumulation of total anthocyanins, as well as greater berry color development, regardless of the application method, directed to the canopy of the vines or only to the bunches. As a result, the new formulation of ACC at concentrations of 75 g to 100 g 100 L^−1^ is a novel tool to stimulate the anthocyanins accumulation and berry color development in ‘Benitaka’ table grapes grown in subtropical regions without negative impact on bunches or vines.

## 1. Introduction

Table grapes are rich sources of phenolic compounds with antioxidant and anti-inflammatory properties, which are useful in preventing several human diseases [1,2]. These secondary metabolites are present in different parts of the berries, where the skin is rich in anthocyanins, a pigment responsible for the pink, red, and black hues of grapes [3]. Anthocyanins specifically account for the highest percentage of phenolic compounds in red grapes [2], representing an important constituent reserved for sensory attributes and berry coloration [4].

‘Benitaka’ (*Vitis vinifera* L.) is an important table grape grown in subtropical climate regions, which originated from a somatic mutation of the ‘Italia’ grape, its most striking characteristic being the dark red color of the berries [5,6]. However, when colored table grapes are grown in areas with hot summers, high temperatures during maturation can inhibit anthocyanin accumulation and prevent skin color development, negatively impacting the market value of table grapes, provided that skin color is a very important characteristic in product pricing [7,8].

As grapes are non-climacteric fruits, to improve quality attributes, plant growth regulators are used along with other agricultural management practices [9]. Among them, abscisic acid (*S*-ABA) induces increased expression of the *VvMYBA1* protein, which is responsible for regulating the genes involved in the biosynthetic pathway of anthocyanins in colored grapes [10], and when the commercial product (ProTone^®^) is applied exogenously to table grapes, it results in significant gains in berry color without altering fruit ripening [11]. Ethylene (C_2_H_4_) is involved in fruit ripening and senescence that occurs naturally in plants, acting on several aspects of fruit maturation, including the accumulation of anthocyanins [12]. Ethylene is a key hormone that regulates grape ripening by activating genes responsible for fruit softening, sugar accumulation, and color changes. It promotes the breakdown of cell wall components and enhances anthocyanin production in red grapes. Its natural production increases as grapes mature, and it can be artificially applied to accelerate ripening [13]. Ethephon (2-chloroethylphosphonic acid), after being applied exogenously to the bunches, is converted to ethylene and is used in colored table grapes to accelerate their ripening and intensify their color; however, it can cause softening and cracking of the berries [8,14].

1-Aminocyclopropane-1-carboxylic acid (ACC) is a non-protein amino acid that is a direct precursor of ethylene present in plants [15]. It is synthesized by the enzyme ACC synthase from methionine and converted to ethylene by the enzyme ACC oxidase [16], thus playing an important role in the ripening and color development of colored grapes. More recently, a novel formulation for agronomic use was developed, containing ACC as the active ingredient, launched on the market under the trade name Accede^®^, which is currently registered as a plant growth regulator to promote thinning of apple fruits. In previous trials, we determined the effect of this new formulation on the color development of table grapes [17]. The hypothesis of this work is that this new formulation, when applied exogenously, can increase the daily rates of anthocyanin accumulation and, as a consequence, improve the berry skin color.

Thus, the objective of this study was to evaluate the action of the exogenous application of different concentrations of the ethylene precursor ACC through applications directed to the vines or only to the bunches regarding the daily rates of anthocyanin accumulation, berry color index, and ripening attributes of table grapes grown in a subtropical region.

## 2. Results

### 2.1. Application to the Entire Canopy of the Vines

The exogenous application of 1-aminocyclopropane-1-carboxylic acid (ACC) directed to the grapevine canopy promoted an increase in the concentration of total anthocyanins in the berries of the table grape ‘Benitaka’ in all the ACC concentrations evaluated (Figure 1). It was also observed that the development of the anthocyanin concentration in the berries better adjusted to the linear model in the evaluated period until the maturation of the bunches occurred 28 days after the application of the treatments, decreasing that the ACC continuously stimulated the accumulation of anthocyanins in the berries. Furthermore, the bunches treated with ACC received daily increment rates that provided a high accumulation of total anthocyanins; however, the values observed for the different ACC concentrations evaluated (25 g, 50 g, 75 g, 100 g, and 125 g 100 L^−1^) were very similar, especially at 21 and 28 days after application, and resulted in monetary rates that ranged from 0.181 to 0.188 mg g^−1^ of anthocyanins per day. Regarding the application of *S*-ABA (3.2 L ha^−1^), the accumulation of anthocyanins throughout the ripening period of the bunches was like the application of ACC (0.188 mg g^−1^ of anthocyanins per day), surpassing the control treatment (0.122 mg g^−1^ of anthocyanins per day). Thus, it can be stated that for the treatments that received the application of ACC or *S*-ABA, the observed increase rates stood out from the control treatment regarding the accumulation of anthocyanins in the bunches.

The development of the berry color index (CIRG) was also best adjusted to the linear model for all treatments evaluated throughout the maturation of the bunches until the day of harvest, which occurred 28 days after the application of the treatments. All ACC concentrations evaluated stood out regarding the development of CIRG, with emphasis on the highest concentrations, which resulted in an increased rate of 0.150 CIRG per day for both concentrations of 100 g and 125 g 100 L^−1^, while the application of *S*-ABA presented a similar result, with a rate of 0.151 CIRG per day. In contrast, the development of CIRG of the bunches of the control treatment was not very expressive throughout the maturation of the bunches, with an increase rate of 0.095 CIRG per day.

Regarding the evaluations of total anthocyanin concentration and CIRG performed when the bunches reached full maturity, it was possible to verify that all the means of the treatments that received the exogenous application of ACC or *S*-ABA were higher than the control treatment for both variables (Table 1). However, for total anthocyanin concentration, there was no statistical difference between the evaluated ACC concentrations (25 g, 50 g, 75 g, 100 g, and 125 g per 100 L^−1^) and *S*-ABA (3.2 L ha^−1^); however, they differed statistically from the control treatment. Regarding the CIRG of the berries, the bunches that received the application of ACC at concentrations of 75 g, 100 g, and 125 g or *S*-ABA 3.2 L presented higher means than the other treatments, followed by the concentrations of 50 g and 25 g of ACC.

For the development of the soluble solids (SS) and titratable acidity (TA) contents of the ‘Benitaka’ grape berries, it was verified from the regression analysis that there was no effect of the exogenous application of ACC directed to the canopy of the vines (Figure 2), occurring similarly for all the treatments evaluated. There were no differences between the treatments when the bunches reached full maturity (Table 2); therefore, the application of the regulators did not affect these physicochemical characteristics of the berries. Consequently, the berry maturity index (SS/TA) was also not influenced by the application of the treatments.

Regarding berry firmness, no statistical difference was observed between the treatments evaluated (Table 2). Thus, the bunches that received the application of different concentrations of ACC or *S*-ABA did not show a reduction in their firmness, and the force required to break the skin of the berries ranged from 13.77 to 16.17 N.

From the sensory–visual analysis of the color coverage classification system of the bunches (Figure 3), it was found that bunches subjected to the application of ACC or *S*-ABA presented higher color coverage categories more quickly than bunches from the control treatment, which progressed until the end of bunch maturation, 28 days after application of the treatments. During this period, bunches treated with ACC, at concentrations of 50 g to 125 g or with *S*-ABA, presented the highest color categories, differing statistically from the control treatment, which had a score lower than 4 on the color coverage scale. Based on the color coverage characteristics of the bunches at harvest (Figure 4 and Appendix A), it was found that those that received the exogenous application of ACC or *S*-ABA at the beginning of bunch maturation were superior, as they presented greater color coverage and intensity when compared to berries or bunches from the control treatment.

### 2.2. Application Only to the Bunchess

The exogenous application of ACC directed only to the bunches of the ‘Benitaka’ table grapes at the beginning of their ripening, as well as that directed to the canopy of the grapevines, also promoted the accumulation of total anthocyanins in the berries (Figure 5), with high rates of daily anthocyanin increases throughout the ripening of the bunches, mainly when the concentrations of 50 g to 125 g 100 L^−1^ were applied, which, in turn, presented equivalent rates, ranging from 0.181 to 0.190 mg g^−1^ of anthocyanins per day, respectively. The bunches treated with *S*-ABA 3.2 L ha^−1^ also stood out regarding the development of anthocyanin accumulation, like the application of ACC (0.186 mg g^−1^ per day). In contrast, for bunches in the control treatment, the increase was only 0.121 mg g^−1^ of anthocyanins per day, resulting in a less expressive accumulation of anthocyanins at the end of bunch maturation.

For CIRG, a behavior similar to that of total anthocyanin accumulation was observed, in which all ACC concentrations evaluated promoted a high increase in CIRG, mainly at concentrations of 75 g to 125 g 100 L^−1^. The application of *S*-ABA also resulted in a significant increase in CIRG, with rates similar to the application of ACC at the highest concentrations, which ranged from 0.140 to 0.148 CIRG per day. On the other hand, the bunches of the control treatment showed an increase of 0.090 CIRG per day, which resulted in a less expressive CIRG development at the end of bunch maturation.

Regarding the concentration of total anthocyanins and CIRG at the end of bunch maturation, it was observed that the bunches treated with ACC or *S*-ABA presented the highest averages (Table 3). However, there were no differences between the different ACC concentrations regarding the accumulation of total anthocyanins, as well as between the application of *S*-ABA. Thus, all treatments, except the concentration of 25 g 100 L^−1^, differed statistically from the control. Regarding CIRG, the treatments that contained ACC at concentrations of 50 g to 125 g per 100 L^−1^ and *S*-ABA 3.2 L ha^−1^ presented the highest averages, differing from the other treatments. In contrast, ACC at a concentration of 25 g 100 L^−1^ did not differ from the control treatment.

Based on the regression analyses for the development of the soluble solids (SS) content, titratable acidity (TA), and ripening index (SS/TA) of the ‘Benitaka’ table grape berries, it was found that these properties were not influenced by the application of plant growth regulators directed only to the bunches (Figure 6). At the harvest of the bunches, it was found that there were no statistical differences between the evaluated treatments (Table 4); therefore, the application of ACC at the different concentrations tested, as well as that of *S*-ABA, did not influence this characteristic of berry ripening. On the other hand, a small reduction in the TA content was observed for the application of ACC at the concentration of 100 g 100 L^−1^ and *S*-ABA 3.2 L ha^−1^; however, these did not differ statistically from ACC at the concentrations of 50 g, 75 g, and 125 g 100 L^−1^, and the averages ranged from 0.58 to 0.64%. Furthermore, this difference was not reflected in the berry ripening index, which varied between 20 and 22 for all bunches. There were also no differences between treatments regarding berry firmness, i.e., although ACC is an ethylene precursor, the treatments applied did not result in more pronounced berry softening.

Based on the sensory–visual analysis of the bunch color coverage system, it was found that bunches subjected to the application of ACC or *S*-ABA grouped themselves in the higher categories more quickly than bunches from the control treatment (Figure 7). At the end of maturation, 28 days after application of the treatments, all bunches that received application of ACC at concentrations of 50 to 125 g 100 L^−1^ or *S*-ABA were in color category 5, which indicates good color coverage. In contrast, the bunches from the control treatment, in this same period, presented a color category lower than 4, differing statistically from the bunches that received the application of plant regulators.

Therefore, the bunches from the control treatment did not achieve sufficient color coverage, which can be observed in Figure 8 and Appendix A, where the superiority of the bunches treated with ACC or *S*-ABA in relation to the control is evident.

## 3. Discussion

This work was developed with the objective of evaluating the concentration of total anthocyanins, a phenolic compound with antioxidant and anti-inflammatory properties, and the pigment responsible for the red color of grapes, in addition to the color development and chemical properties of the berries of the table grape ‘Benitaka’, in response to the exogenous application of 1-aminocyclopropane-1-carboxylic acid (ACC), directed to the vine canopies, or only to the bunches.

It was observed that ACC applied at the beginning of bunch ripening resulted in a greater accumulation of total anthocyanins and berry color, regardless of the application method. This can be attributed to the effects of ACC, which, when applied exogenously to vines, is converted into ethylene in plants via the Yang cycle from the methionine since ACC is its immediate precursor [15,18]. Ethylene, in turn, stimulates the expression of several enzymes that are directly related to anthocyanin biosynthesis, such as phenylalanine ammonia-lyase (PAL) and UDP-glucose (UFGT) [11,19].

Therefore, bunches that received ACC application tend to accumulate anthocyanins in the berries more quickly. In addition, a greater advance in the development of anthocyanin concentration can be observed 14 days after the application of the treatments, and this pattern was observed for both forms of ACC application evaluated (Figure 1 and Figure 5). This same pattern was also observed by Shahab et al. [20] and Ribeiro et al. [21] when studying the effect of exogenous application of *S*-ABA in ‘Benitaka’ and ‘Rubi’ table grapes. This may be linked to the higher sugar content in the berries during this period, which may have provided a substrate to initiate the production of secondary metabolites, such as anthocyanins, responsible for the red color of the skin [22,23]. This corroborates Keller [24], who reported that the onset of expression of genes involved in anthocyanin synthesis occurs when the SS content ranges from 9 to 10 °Brix, as observed in this study.

In general, red grapes with high total anthocyanin content in the skin have a more intense color than grapes with low anthocyanin content [14]. However, this relationship between pigment content and berry color characteristics may not be linear, and a large accumulation of pigment content may have little effect on berry color [14,25]. However, in this study, it was observed that skin color development followed a pattern equivalent to that observed for total anthocyanin accumulation.

CIRG is an important instrumental variable in evaluating the effect of ACC on grape color since it has a high correlation with the visual color of the berries [26]. According to Carreño et al. [26], at the end of the ripening of the bunches, the berries of the control treatment had a red color, while the berries treated with ACC or *S*-ABA had a dark red to red–black color. This difference demonstrates the effectiveness of the treatments in intensifying the color of the grape. ‘Benitaka’ is a *V. vinifera* grape; therefore, its appearance and color are crucial to obtaining the best marketing price since consumers are more attracted to the intense red color of the berries [27]. Furthermore, normally, the more intense the color of the berries, the higher the content of phenolic compounds, and the presence of these compounds presents benefits to human health with high antioxidant and anti-inflammatory activity [28,29,30]. Therefore, in addition to being important for the visual characteristics of the bunches, grapes with greater color intensity have a greater beneficial effect on human health [27,31].

It has been demonstrated that applications of *S*-ABA in véraison increase the anthocyanin content as well as the color of berries of several grape cultivars [6,9,14,20,21,28,31]. This is attributed to the effects of *S*-ABA on the expression of genes related to anthocyanin biosynthesis [32,33,34]. However, the effects of the exogenous application of the recent formulation containing ACC developed for use in agriculture for this same purpose were not yet known. It was verified in this work that ACC can be a new alternative to overcome the problems of coverage and color intensity in table grapes grown in a subtropical climate region. It is noteworthy that with only one application of ACC at concentrations of 50 g to 125 100 L^−1^, the result was the same as with two applications of *S*-ABA 3.2 L ha^−1^ and may be more advantageous due to the lower costs involved.

There was no influence of exogenous ACC application on the physicochemical characteristics of the berries, and similarly, no effect for *S*-ABA was observed on the physicochemical characteristics of the grapes ‘Flame Seedless’ [35], ‘Crimson Seedless’ [23,36], ‘Sovereign Coronation’ [37], ‘BRS Melodia’ [31], as well as for ‘Benitaka’ grape [6], and also, no effect of ethephon was observed for ‘Crimson Seedless’ [25], ‘Rubi’ [38], ‘Cabernet Sauvignon’ [39], and ‘Thompson Seedless’ [40]. The physicochemical characteristics of the fruits are not normally influenced by the plant growth regulators but rather by environmental conditions and cultural practices during bunch ripening [6,22,23]. Therefore, although ACC is an ethylene precursor, regardless of the form of application, its exogenous application does not alter bunch ripening.

Regarding berry firmness, despite evidence reported by several authors that ethephon can result in berry softening [14,41,42,43], the exogenous application of ACC, even at its highest concentration (125 g 100 L^−1^), did not cause berry softening. Although both regulators act on ethylene biosynthesis, this conjugation occurs differently. Ethephon applied exogenously to plants rapidly releases ethylene into the tissues, which will play its role in fruit ripening [44]. This rapid increase in ethylene in the tissues may be responsible for causing cell wall weakening and cracking in the berries. In contrast, ethylene conversion from exogenous ACC application is not immediate and will only occur in tissues in which ACC oxidase is present, and the tissue is sensitive to ethylene for any response to exogenous ACC to occur [45]. In addition, ACC can also be conjugated to 1-malonyl-ACC, ϒ-glutamyl-ACC, or jasmonyl-ACC; these derivatives are a biochemical bypass to regulate the fraction of ACC available for ethylene production [46]. Each step of this pathway is highly regulated to control the development and progression of ethylene-dependent phenomena [15,16,47]; this controlled increase in ethylene content did not result in a reduction in berry firmness.

Firmness loss due to *S*-ABA application was observed in ‘Flame Seedless’, ‘Redglobe’, ‘Crimson Seedless’ [7,14,43], and ‘Benitaka’ grapes [19]; however, it does not corroborate the results obtained in this study as well as those found in Olivares et al. [23]; Ferrara et al. [40] and Lurie et al. [48] in ‘Crimson Seedless’. Thus, berry softening induced by *S*-ABA is not consistent. It is worth noting that firmness is a relevant characteristic for the postharvest handling of grape bunches destined for the fresh fruit market, directly influencing the useful life of the bunches [25,49]. However, no softening of the berries was observed due to the application of treatments at the beginning of the ripening of the bunches.

In this study, it was also observed that exogenous ACC improved both the intensity and uniformity of fruit color. The color coverage of the control treatment bunches showed the lowest color coverage; in addition, these untreated bunches would probably never reach the adequate color, while the ACC treatment provided bunches with a more intense and uniform color. More intense colors and greater coverage in the berries were also observed in ‘Benitaka’ and ‘Rubi’ table grapes after exogenous application of *S*-ABA [6,8].

Regarding the forms of application of ACC evaluated, whether applied to the vine or only to the bunches, both presented similar performance for all variables evaluated. It is worth mentioning that, when applied only to the bunches using a backpack sprayer, it is a laborious and time-consuming task since all parts of the bunches need to be completely covered; however, the application volume is smaller than when applied to the entire canopy of the vines, reducing production costs and making it suitable for small properties, where a single operator performs this application. On the other hand, the application of ACC to the entire canopy of the vines using a tractor-mounted air jet sprayer is very effective in achieving complete coverage, making it ideal for application in large areas.

Finally, further studies are needed, including exogenous applications of different plant growth regulators to intensify the color of grape clusters without causing a negative impact on the berries. One possibility to achieve this objective is the association of plant growth regulators, which act on different metabolic routes in the plant, as the combination of reduced concentrations of ACC, *S*-ABA, and ethephon in one application, aiming to increase their effectiveness to improve berry color.

## 4. Materials and Methods

### 4.1. Location of Experiments and Grape Cultivar Evaluated

This study was conducted in a commercial area located in the municipality of Cambira, PR, coordinates 23°35′27″ S 51°34′14″ W, elevation of 820 m, in a vineyard of 2-year-old ‘Benitaka’ grapevine (*Vitis vinifera* L.), grafted onto ‘IAC 766 Campinas’ rootstock, with the vines trained on overhead trellises system spaced at 2.5 × 3.5 m and protected under a black screen with 18% shading.

The climate of the region is classified as humid subtropical (Cfa) by Köppen, with an average winter temperature below 18 °C, an average summer temperature above 22 °C, and 1600 mm of precipitation, which occurs mainly in the summer [50].

### 4.2. Experimental Design and Treatments

The effect of the application of 1-aminocyclopropane-1-carboxylic acid—ACC, in different concentrations and forms of application, as well as abscisic acid—*S*-ABA, on the color development of ‘Benitaka’ grape berries during the 2022 harvest was evaluated.

The experimental design was randomized blocks consisting of seven treatments and four replicates, with each plot consisting of one vine. The commercial product Accede^®^ (Valent Biosciences, Libertyville, IL, USA) with 40% of the active ingredient (400 g kg^−1^, water-soluble granules, SG) was used as the ACC source, and the treatments included concentrations of 0, 25, 50, 75, 100, and 125 g 100 L^−1^ of the commercial product, as well as the standard concentration of *S*-ABA for comparisons, using the commercial product ProTone^®^ (Valent Biosciences, Libertyville, IL, USA) with 10% of the active ingredient (100 g L^−1^, soluble concentrate, SL) at 3.2 L ha^−1^.

The exogenous application of ACC was carried out in a single application, in two ways in independent trials [17]: to the entire canopy of the vines, at an estimated application volume of 1000 L ha^−1^, or only to the bunches, at an estimated application volume of 300–400 L ha^−1^, while the *S*-ABA was applied only to the bunches in two applications [6]. The application of all treatments was carried out early in the morning (air temperature around 23 °C) at the beginning of the berry ripening (véraison), when the soluble solids content reached around 7.5°Brix. For the exogenous application of *S*-ABA, the second application was carried out 7 days after the first one [20]. To promote better absorption of the growth plant regulators, 0.5 mL L^−1^ of non-ionic adhesive spreader BreakThru^®^ (Evonik, Brazil) was added to the spray of all treatments.

### 4.3. Assessments

The evaluations were performed weekly, from the beginning of the treatment applications until the full maturation of the bunches. For this purpose, five bunches were previously marked per plot. In each evaluation, 10 berries were sampled per plot, with two berries from each bunch previously marked.

Thus, the following variables were assessed weekly: total anthocyanin concentration, berry color index (CIRG), soluble solids (SS) and titratable acidity (TA), maturation index or ratio (SS/TA), in addition to visual evaluation of the color coverage of the bunches, while firmness was evaluated at harvest.

To evaluate the total anthocyanin concentration of the berries, 10 berries per plot were evaluated using the methodology adapted from Peppi, Fidelibus, and Dokoozlian [7]. The results were expressed in mg of malvidin-3-glucoside (mal-3-glu) per g of skin (mg g^−1^).

The evaluation of the berry color was performed by using a colorimeter (Konica-Minolta, CR 10 Plus), analyzing the 10 berries of each plot. The values obtained during the determination of this variable were expressed in the coordinates of the CIELab standard. From these values, the berry color index (CIRG) was calculated by the equation: CIRG = 180 − *h^o^*/(*L** + *C**) [51].

Berry firmness was assessed using an Instrutherm DD-200 digital penetrometer, analyzing 10 berries per plot. Each berry was placed individually under the base of the penetrometer and manually compressed in its equatorial diameter with a cylindrical probe until the skin broke, and the results were expressed in N.

Regarding the analysis of SS and TA contents and SS/TA, 10 berries from each plot were evaluated and crushed to extract the juice. The SS content was obtained using a digital refractometer (Model PAL-1, Atago), and the results were expressed in °Brix. The TA was determined by the titration method of the Adolfo Lutz Institute [52]. Furthermore, the maturation index was obtained through the relationship between SS and TA (SS/TA).

The color coverage analysis of the bunches was performed through sensory–visual evaluation carried out by 10 trained panelists. For this purpose, the classification system composed of 6 categories or classes of color coverage of the bunches for ‘Benitaka’ grapes described by Aguiar et al. [17] was used: Class 1: 100% green color coverage; Class 2: 1–25% red color coverage; Class 3: 25–50% red color coverage; Class 4: 50–75% red color coverage; Class 5: 75–100% red color coverage; and Class 6: 100% intense red or red–purple color coverage.

The data were subjected to analysis of variance (ANOVA), and when significant, the means were compared by Tukey’s test (*p* < 0.05) using the R-Studio software. The weekly means of the accumulation of total anthocyanins, CIRG, SS, TA, and SS/TA of the berries were adjusted by means of polynomial regressions (*p* < 0.05), and the fit of the models was determined from the largest coefficients of determination (*R*^2^). Finally, the daily rates of increase in these variables were determined for each treatment according to the adjusted regression equations.

## 5. Conclusions

A single exogenous application of the novel formulation of ACC or two applications of *S*-ABA resulted in daily growth rates that provided a high accumulation of total anthocyanins as well as greater color development in the berries, regardless of the application method, directed to the vines or only to the bunches. The new formulation of ACC at concentrations of 75 g to 100 g 100 L^−1^ is a novel tool to stimulate the anthocyanins accumulation and berry color development in ‘Benitaka’ table grapes grown in subtropical regions, without negative impact on bunches or vines.

## Figures and Tables

**Figure 1 plants-14-01058-f001:**
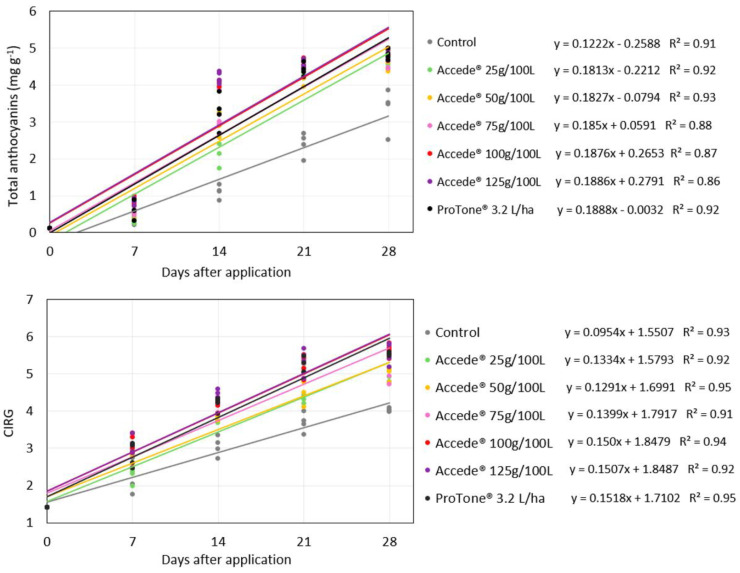
Development of total anthocyanin concentration and berry color index—CIRG of ‘Benitaka’ table grape (*Vitis vinifera* L.) at 0, 7, 14, 21, and 28 days after treatments with 1-aminocyclopropane-1-carboxylic acid—Accede^®^ and abscisic acid—ProTone^®^ with application to the entire canopy of the vines of bunch maturation.

**Figure 2 plants-14-01058-f002:**
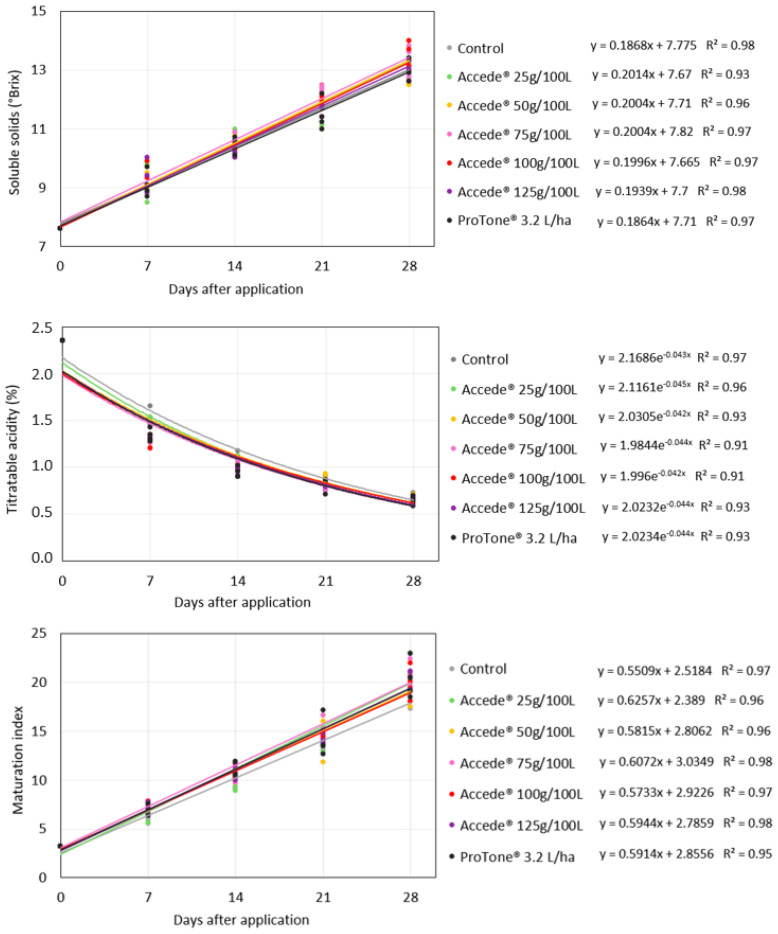
Development of soluble solids content (°Brix), titratable acidity, and ripening index of ‘Benitaka’ table grape berries (*Vitis vinifera* L.) at 0, 7, 14, 21, and 28 days after application of different treatments with 1-aminocyclopropane-1-carboxylic acid—Accede^®^ and abscisic acid—ProTone^®^ applied to the entire canopy of the vines at véraison.

**Figure 3 plants-14-01058-f003:**
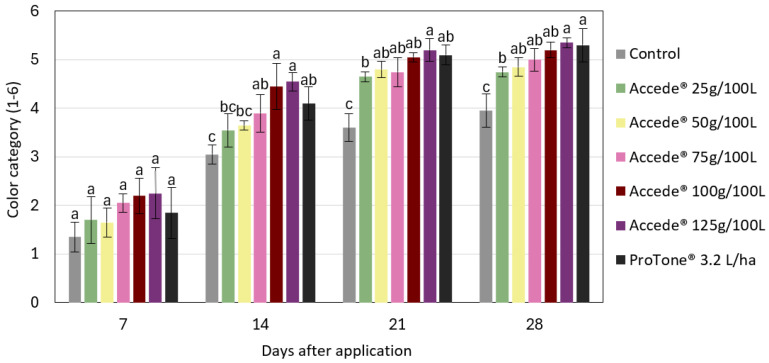
Color coverage categories of ‘Benitaka’ grape (*Vitis vinifera* L.) at 7, 14, 21, and 28 days after application of different treatments with 1-aminocyclopropane-1-carboxylic acid—Accede^®^ and abscisic acid—ProTone^®^ applied to the entire canopy of the vines at véraison. Means followed by the same letters within each sampling time do not differ by Tukey’s test (*p* < 0.05%).

**Figure 4 plants-14-01058-f004:**
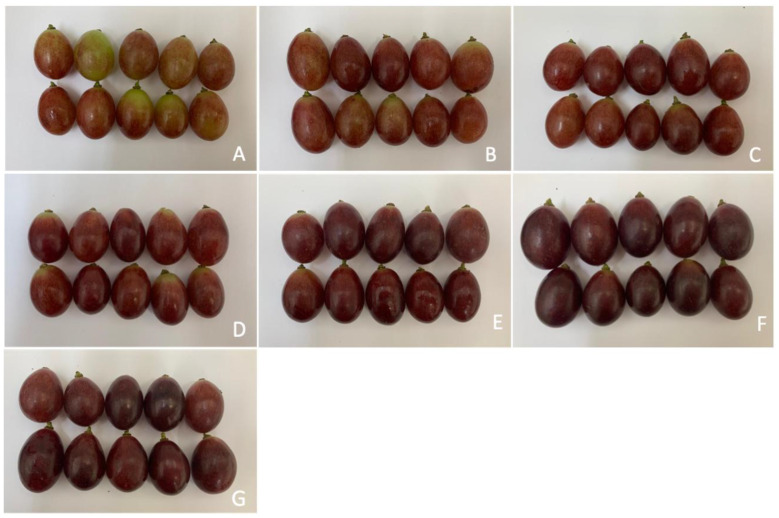
Berries of ‘Benitaka’ table grape (*Vitis vinifera* L.) subjected to different treatments with 1-aminocyclopropane-1-carboxylic acid—Accede^®^ and abscisic acid—ProTone^®^ applied to the entire canopy of the vines at véraison. Control (**A**); Accede^®^ 25; 50; 75; 100; 125 g 100 L ^−1^ (**B**–**F**); ProTone^®^ 3.2 L ha^−1^ (**G**).

**Figure 5 plants-14-01058-f005:**
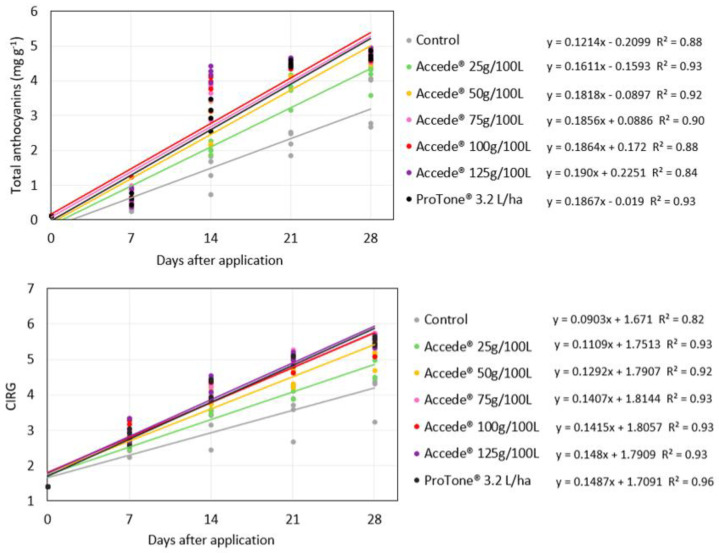
Development of total anthocyanin concentration and berry color index—CIRG of ‘Benitaka’ table grape (*Vitis vinifera* L.) at 0, 7, 14, 21, and 28 days after treatments with 1−aminocyclopropane−1−carboxylic acid—Accede^®^ and abscisic acid—ProTone^®^ applied only to the bunches at véraison.

**Figure 6 plants-14-01058-f006:**
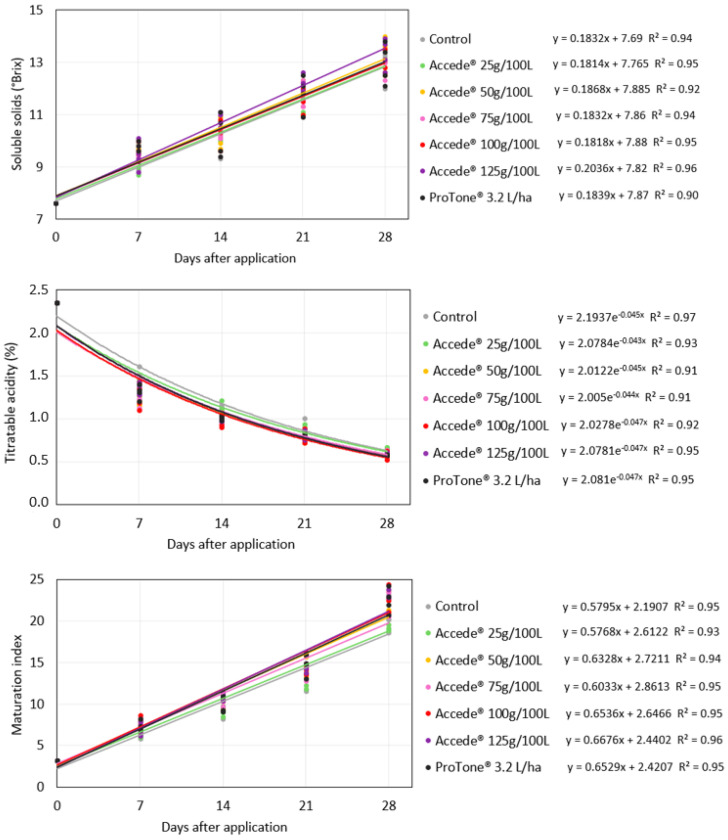
Development of soluble solids content (°Brix), titratable acidity and ripening index of ‘Benitaka’ table grape berries (*Vitis vinifera* L.) at 0, 7, 14, 21, and 28 days after application of different treatments with 1-aminocyclopropane-1-carboxylic acid—Accede^®^ and abscisic acid—ProTone^®^ applied only to the bunches at véraison.

**Figure 7 plants-14-01058-f007:**
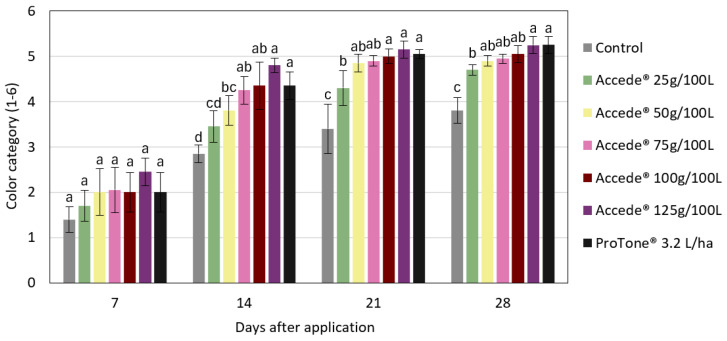
Color coverage categories of ‘Benitaka’ grape (*Vitis vinifera* L.) at 7, 14, 21, and 28 days after application of different treatments with 1-aminocyclopropane-1-carboxylic acid—Accede^®^ and abscisic acid—ProTone^®^ applied only to the bunches at véraison. Same letters within each sampling time do not differ by Tukey’s test (*p* < 0.05%).

**Figure 8 plants-14-01058-f008:**
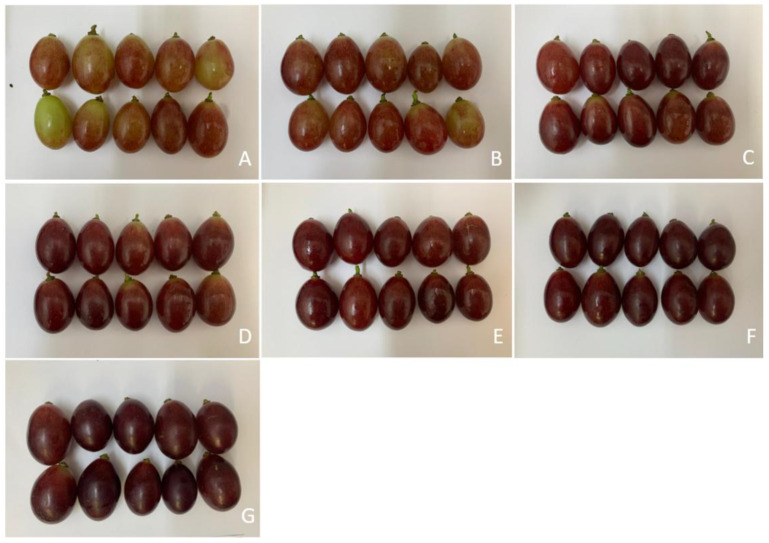
Berries of the table grape ‘Benitaka’ (*Vitis vinifera* L.) subjected to different treatments with 1-aminocyclopropane-1-carboxylic acid—Accede^®^ and abscisic acid—ProTone^®^ applied only to the bunches at véraison. Control (**A**); Accede^®^ 25; 50; 75; 100; 125 g 100 L ^−1^ (**B**–**F**); ProTone^®^ 3.2 L ha^−1^ (**G**).

**Table 1 plants-14-01058-t001:** Total anthocyanins and berry color index—CIRG of ‘Benitaka’ table grape (*Vitis vinifera* L.) subjected to different treatments with 1-aminocyclopropane-1-carboxylic acid—Accede^®^ and abscisic acid—ProTone^®^ applied to the entire canopy of the vines at véraison.

Treatments (Concentrations per 100 L or ha)	Total Anthocyanins (mg mal-3-glu g^−1^)	Berry Color Index (CIRG)
Control	3.34 b	4.03 c
Accede^®^ 25 g	4.53 a	4.99 b
Accede^®^ 50 g	4.62 a	5.01 b
Accede^®^ 75 g	4.62 a	5.26 ab
Accede^®^ 100 g	4.83 a	5.65 a
Accede^®^ 125 g	4.84 a	5.55 ab
ProTone^®^ 3.2 L	4.82 a	5.53 ab
F	16.43 **	18.19 **
CV (%)	5.79	5.08

Means followed by the same letters in the columns do not differ by Tukey’s test (*p* < 0.05%). **: significant (*p* < 0.01). CV: coefficient of variation.

**Table 2 plants-14-01058-t002:** Soluble solids—SS; titratable acidity—TA; maturation index—SS/TA and firmness of ‘Benitaka’ table grape berries (*Vitis vinifera* L.) subjected to different treatments with 1-aminocyclopropane-1-carboxylic acid—Accede^®^ and abscisic acid—ProTone^®^ applied to the entire canopy of the vines at véraison.

Treatments (Concentrations per 100 L or ha)	SS (°Brix)	TA (%)	SS/TA	Firmness (N)
Control	12.85	0.69	18.79	16.17
Accede^®^ 25 g	13.35	0.64	20.99	15.78
Accede^®^ 50 g	13.30	0.66	20.04	15.28
Accede^®^ 75 g	13.30	0.65	20.53	14.28
Accede^®^ 100 g	13.35	0.67	19.88	14.30
Accede^®^ 125 g	13.18	0.64	20.51	14.05
ProTone^®^ 3.2 L	12.95	0.64	20.26	13.77
F	0.75 ^ns^	1.20 ^ns^	0.80 ^ns^	2.25 ^ns^
CV (%)	3.58	5.11	7.76	8.4%

Means followed by the same letters in the columns do not differ by Tukey’s test (*p* < 0.05%). ^ns^: not significant. CV: coefficient of variation.

**Table 3 plants-14-01058-t003:** Total anthocyanins and berry color index—CIRG of ‘Benitaka’ table grape (*Vitis vinifera* L.) subjected to different treatments with 1-aminocyclopropane-1-carboxylic acid—Accede^®^ and abscisic acid—ProTone^®^ applied only to the bunches at véraison.

Treatments (Concentrations per 100 L or ha)	Total Anthocyanins (mg mal-3-glu g^−1^)	Berry Color Index (CIRG)
Control	3.51 b	4.10 c
Accede^®^ 25 g	4.12 ab	4.62 bc
Accede^®^ 50 g	4.57 a	5.15 ab
Accede^®^ 75 g	4.68 a	5.39 a
Accede^®^ 100 g	4.75 a	5.36 a
Accede^®^ 125 g	4.81 a	5.51 a
ProTone^®^ 3.2 L	4.71 a	5.52 a
F	7.14 **	11.42 **
CV (%)	8.56%	6.22%

Means followed by the same letters in the columns do not differ by Tukey’s test (*p* < 0.05%). **: significant (*p* < 0.01). CV: coefficient of variation.

**Table 4 plants-14-01058-t004:** Soluble solids—SS; titratable acidity—TA; maturation index—SS/TA, and firmness of ‘Benitaka’ table grape berries (*Vitis vinifera* L.) subjected to different treatments with 1-aminocyclopropane-1-carboxylic acid—Accede^®^ and abscisic acid—ProTone^®^ applied only to the bunches at véraison.

Treatments (Concentrations per 100 L or ha)	SS (°Brix)	TA (%)	SS/TA	Firmness (N)
Control	12.87	0.64 a	20.02	16.07
Accede^®^ 25 g	12.80	0.64 a	20.16	16.82
Accede^®^ 50 g	13.22	0.60 ab	22.19	12.87
Accede^®^ 75 g	13.05	0.61 ab	21.31	14.54
Accede^®^ 100 g	13.00	0.58 b	22.66	12.04
Accede^®^ 125 g	13.33	0.59 ab	22.58	13.82
ProTone^®^ 3.2 L	12.95	0.58 b	22.43	14.09
F	0.38 ^ns^	5.46 **	2.07 ^ns^	1.55 ^ns^
CV (%)	4.63%	3.87%	7.29%	18.91%

Means followed by the same letters in the columns do not differ by Tukey’s test (*p* < 0.05%). **: significant (*p* < 0.01). ^ns^: not significant. CV: coefficient of variation.

## Data Availability

Original data are available upon request.

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
