# Peer review of "Application of a Novel Formulation of 1-Aminocyclopropane-1-carboxylic Acid (ACC) to Increase the Anthocyanins Concentration in Table Grape Berries"

_plants, 2025, doi:10.3390/plants14071058_

Round 1

Reviewer 1 Report

Comments and Suggestions for Authors

The manuscript reports the exogenous application of ACC in anthocyanin accumulation in table grapes. This work presents valuable insights and address a significant result in the field of horticultural sciences. However, the authors need to provide some data and need to clarify some important issues. For example, in the results, ethylene and respiration data are missing in this study. Additionally, the spelling and typing errors should be corrected. Moreover, the plagiarism of this study is 41% and it is not acceptable.

Lines 53-55: Grape is non-climacteric. How ethylene involved fruit ripening in grape? Please explain.

Introduction: There has no information about ‘ProTone’. Add the information for this treatment.

Poor in figure quality: The author should improve the quality and resolution of all figures.

Results: the authors used the ethylene precursor ACC treatments in this study, but no results of ethylene and respiration were reported. Provide the results and discussed properly.

In Figures 2, 5: The results of SS, TA, SS/TA, firmness, anthocyanin, CIRG are presented. But in Table 1, 2: The results of SS, TA, SS/TA, firmness, anthocyanin, CIRG are presented. These data are the harvest data? Table 3: The results of anthocyanin and CIRG are presented again? Why?

Lines 411-413: This study mainly focused on anthocyanin improvement in the grape. The authors must describe the determination method in details.

Lines 427-428: The method for maturation index was unclear. Provide more detail information.

Line 438: Correct the spelling to ‘TA’, not AT.

Conclusion: The authors stated that ‘ACC is a novel strategy to trigger anthocyanin accumulation and color development of table grapes without negative impacts on bunches”. This is 100% success. However, as my suggestion, the authors should describe “limitation of the present study” because the authors used only small sample size in this study. Additionally, this study conducted on a single year and spraying treatments highly depend on weather conditions and can vary year by year. Moreover, further researches should be described as a suggestion for future researches. 

Author Response

Reviewer #1:

Comments and Suggestions for Authors

The manuscript reports the exogenous application of ACC in anthocyanin accumulation in table grapes. This work presents valuable insights and address a significant result in the field of horticultural sciences. However, the authors need to provide some data and need to clarify some important issues. For example, in the results, ethylene and respiration data are missing in this study. Additionally, the spelling and typing errors should be corrected.

Response: Dear reviewer, thank you very much for your valuble considerations. The text was subjected to an English Editing to solve spelling and typos, and the similarity index was also reduced.

Lines 53-55: Grape is non-climacteric. How ethylene involved fruit ripening in grape? Please explain.

Response: This is information was added to the text (see L60-66).

Introduction: There has no information about ‘ProTone’. Add the information for this treatment.

Response: Information about ProTone was added to this section (see L44).

Poor in figure quality: The author should improve the quality and resolution of all figures.

Response: Images with high resolution will be provided to Plants editorial office.

Results: the authors used the ethylene precursor ACC treatments in this study, but no results of ethylene and respiration were reported. Provide the results and discussed properly.

Response: Indeed, in our trials, we assessed the impact of this new formulation of ACC developed for agricultural use on anthocyanins accumulation and color index only. We agree that ethylene and respiration assessment are important, but they were not included. This valuble remark will be considered in future trials.

In Figures 2, 5: The results of SS, TA, SS/TA, firmness, anthocyanin, CIRG are presented. But in Table 1, 2: The results of SS, TA, SS/TA, firmness, anthocyanin, CIRG are presented. These data are the harvest data?

Response: Yes, Table 1 and 2 show the means observed during the harvest of grapes as mentioned in the Title.

Table 3: The results of anthocyanin and CIRG are presented again? Why?

Response: Indeed, the results shown in Tables 1 and 2 are related to the exogenous application to the entire canopy of the vines, while the results are related to the application to the bunches only.

Lines 411-413: This study mainly focused on anthocyanin improvement in the grape. The authors must describe the determination method in details.

Response: The determination method to assess anthocyanin in grape berries has been described in several publications, including ours. For this reason, to reduce the similarity index, we rather to cite the original work in which this methodology is well described:

Peppi, M.C.; Fidelibus, M.W.; Dokoozlian, N. Abscisic Acid Application Timing and Concentration Affect Firmness, Pigmen-tation, and Color of `Flame Seedless' Grapes. HortScience 2006, 41, 1440-1445.  https://doi.org/10.21273/HORTSCI.41.6.1440

Lines 427-428: The method for maturation index was unclear. Provide more detail information.

Response: The maturation index was calculated by using the relation SS/TA (see L425-427).

Line 438: Correct the spelling to ‘TA’, not AT.

Response: Changes were made as requested.

Conclusion: The authors stated that ‘ACC is a novel strategy to trigger anthocyanin accumulation and color development of table grapes without negative impacts on bunches”. This is 100% success. However, as my suggestion, the authors should describe “limitation of the present study” because the authors used only small sample size in this study. Additionally, this study conducted on a single year and spraying treatments highly depend on weather conditions and can vary year by year. Moreover, further researches should be described as a suggestion for future researches.

Response: See our remarks in L381-386.

Reviewer 2 Report

Comments and Suggestions for Authors

This manuscript reports that the anthocyanin content in table grape berries increases with the treatment of ACC, a precursor of ethylene. Although this manuscript has great practical significance, the mechanism by which anthocyanin increased has not been studied at all, and the content is extremely superficial. Therefore, this manuscript does not fit the orientation of plants, which focuses on plant science. Even in a horticultural journal, experimental data that provides insight into the mechanism by which ACC treatment increases the anthocyanin content, such as gene expression or enzyme activity related to the biosynthesis of anthocyanin, is necessary. This manuscript uses Accede, which contains ACC as its main component, but does not state the concentration of ACC itself. In scientific papers, pure reagents with guaranteed purity should be used as a general rule. As mentioned above, this manuscript does not provide results worthy of publication in an international journal, and therefore this manuscript is not appropriate for publication in plants.

Author Response

Reviewer #2:

Comments and Suggestions for Authors

This manuscript reports that the anthocyanin content in table grape berries increases with the treatment of ACC, a precursor of ethylene. Although this manuscript has great practical significance, the mechanism by which anthocyanin increased has not been studied at all, and the content is extremely superficial. Therefore, this manuscript does not fit the orientation of plants, which focuses on plant science. Even in a horticultural journal, experimental data that provides insight into the mechanism by which ACC treatment increases the anthocyanin content, such as gene expression or enzyme activity related to the biosynthesis of anthocyanin, is necessary. This manuscript uses Accede, which contains ACC as its main component, but does not state the concentration of ACC itself. In scientific papers, pure reagents with guaranteed purity should be used as a general rule. As mentioned above, this manuscript does not provide results worthy of publication in an international journal, and therefore this manuscript is not appropriate for publication in plants.

Response: Dear reviewer, thanks for your relevant remarks. We would like to inform you that the commercial product used (Accede) contains 40% of the active ingredient (ACC), as mentioned in Material and Methods section. This novel formulation was developed for agricultural use. As pure reagents are available, they are not allowed to be used in agriculture. Our manuscript is original, and we demonstrated the impact of this new formulation to improve the color of grape berries grown in subtropics.

Reviewer 3 Report

Comments and Suggestions for Authors

11 March, 2025

Manuscript ID: plants-3529701

Title of the manuscript:  Exogenous Application of a New Formulation of 1-Aminocyclopropane-1-carboxylic Acid (ACC) to Increase the Anthocyanins Concentration in Table Grape Berries

In the study, information about “exogenous application of a new formulation of 1-aminocyclopropane-1-carboxylic acid (ACC) to increase the anthocyanins concentration in table grape berries” is provide. I have completed the evaluation of the manuscript you sent me for review. The manuscript is detailed, original and well written. Homogeneous coloring is a big problem in table grapes. Table grapes that are not homogeneously colored are sold at low prices in the market. Therefore, coloring is encouraged by manipulating growth regulators. The study focused on solving the coloring problem in a table grape variety. I think manuscript will contribute to filling this gap in knowledge and will attract the attention of readers. The research question and hypothesis of the study, is missing, should be added. My other suggestions were shown on the annotated PDF file.

After MINOR corrections, the manuscript can be accepted for publication in PLANTS.

Note: My suggestions were shown on the annotated PDF file.

With my best regards

Author Response

Reviewer #3:

Comments and Suggestions for Authors

11 March, 2025

Manuscript ID: plants-3529701

Title of the manuscript:  Exogenous Application of a New Formulation of 1-Aminocyclopropane-1-carboxylic Acid (ACC) to Increase the Anthocyanins Concentration in Table Grape Berries

In the study, information about “exogenous application of a new formulation of 1-aminocyclopropane-1-carboxylic acid (ACC) to increase the anthocyanins concentration in table grape berries” is provide. I have completed the evaluation of the manuscript you sent me for review. The manuscript is detailed, original and well written. Homogeneous coloring is a big problem in table grapes. Table grapes that are not homogeneously colored are sold at low prices in the market. Therefore, coloring is encouraged by manipulating growth regulators. The study focused on solving the coloring problem in a table grape variety. I think manuscript will contribute to filling this gap in knowledge and will attract the attention of readers. The research question and hypothesis of the study, is missing, should be added. My other suggestions were shown on the annotated PDF file.

After MINOR corrections, the manuscript can be accepted for publication in PLANTS.

Note: My suggestions were shown on the annotated PDF file.

With my best regards

Response: Dear reviewer, thank you very much. All you comments and suggestions were accepted and incoporporated to the manuscript.

Round 2

Reviewer 1 Report

Comments and Suggestions for Authors

The authors addressed all my comments.

Thanks.

Reviewer 2 Report

Comments and Suggestions for Authors

I found some improvements in the revised manuscript, but it is by no means publishable. I have decided that it should not be published unless new data is added and it is compiled as a completely separate manuscript.